# Spatiotemporal Variation in Air Pollution Characteristics and Influencing Factors in Ulaanbaatar from 2016 to 2019

**Suriya** [1,2,*], **Narantsogt Natsagdorj** [2], **Aorigele** [3], **Haijun Zhou** [4] **and Sachurila** [5]

1 College Chemistry and Chemical Engineering, Hulunbuir University, Hulunbuir 021008, China
2 Department of Chemistry, School of Mathematics and Natural Science, Mongolian National University of Education, Ulaanbaatar 210648, Mongolia; narantsogt@msue.edu.mn
3 Academic Affairs Office, Hulunbuir University, Hulunbuir 021008, China; aorigele@hlbec.edu.cn
4 College of Geographical Sciences, Inner Mongolia Normal University, Hohhot 010022, China; hjzhou@imnu.edu.cn
5 School of Resources, Environment and Architectural Engineering, Chifeng University, Chifeng 024000, China; scrl@sina.com
* Correspondence: suriyaa@hlbec.edu.cn; Tel./Fax: +976-9581-9724

**Abstract:** Ambient air pollution is a global environmental issue that affects human health. Ulaanbaatar (UB), the capital of Mongolia, is one of the most polluted cities in the world, and it is of great importance to study the temporal and spatial changes in air pollution in this city, along with their influencing factors. To understand the characteristics of atmospheric pollutants in UB, the contents of $PM_{10}$, $PM_{2.5}$, $SO_2$, $NO_2$, CO, and $O_3$, as well as their influencing factors, were analyzed from data obtained from automatic air quality monitoring stations. These analyses yielded six major findings: (1) From 2016 to 2019, there was a total of 883 pollution days, and $PM_{2.5}$ and $PM_{10}$ were the primary pollutants on 553 and 351 of these days, respectively. The air pollution was dominated by $PM_{10}$ in spring and summer, affected by both $PM_{2.5}$ and $PM_{10}$ in autumn, and dominated by $PM_{2.5}$ in winter. (2) Compared with 2016, the number of days with good air quality in UB in 2019 increased by 45%, and the number of days with unhealthy or worse levels of pollution decreased by 56%, indicating that the air quality improved year by year. (3) From 2016 to 2019, the annual average $PM_{2.5}/PM_{10}$ ratio dropped from 0.55 to 0.45, and the proportion of $PM_{2.5}$ in particulate matter decreased year by year. The PM concentration and $PM_{2.5}/PM_{10}$ ratio were highest in winter and lowest in summer. When comparing the four-season averages, the average $PM_{2.5}$ concentration decreased by 89% from its highest level, and the $PM_{10}$ concentration decreased by 67%, indicating stronger seasonal differences in $PM_{2.5}$ than in $PM_{10}$. (4) The hourly changes in PM concentration showed a bimodal pattern, exhibiting a decrease during the day and a slight increase in the afternoon due to temperature inversion, so the $PM_{2.5}/PM_{10}$ ratio increased at night in all four seasons. The PM concentration during the heating season was significantly higher than that in the non-heating season, indicating that coal-fired heating was the main cause of air pollution in UB. (5) Sand dust and soot were the two main types of pollution in UB. (6) Correlation analysis and linear fitting analysis showed that $PM_{2.5}$ and $PM_{10}$ caused by coal-firing had an important impact on air quality in UB. Coal combustion and vehicle emissions with $SO_2$, $NO_2$, and CO as factors made large contributions to $PM_{2.5}$.

**Keywords:** air quality; particulate matter; gaseous pollutants; impact factors; correlation analysis

## 1. Introduction

Ambient air pollution is a global environmental problem [1,2] that has serious negative impacts on climate change, visibility, and human health [3–5]. For example, about 4.2 million people worldwide died of heart disease, stroke, lung disease, and chronic respiratory disease caused by air pollution in 2017 [6]. Human factors have a significant impact on urban air pollution. Industrialization, urbanization, modernization of transportation,

and increased energy consumption often lead to increased emissions of air pollutants, resulting in degraded urban air quality [7–9]. In recent decades, several countries and regions have formulated and implemented various air pollution prevention and control measures, achieving varying degrees of air quality improvement [10–12].

One such country is Mongolia, which is rich in mineral resources, with large reserves of coal, copper, and gold. Despite its vast land area and sparse population, it is one of the most polluted countries in the world. According to the "2018–2020 World Air Quality Report" published on the IQAir website (https://www.iqair.com/), in 2018, 2019, and 2020 Mongolia ranked sixth, third, and fourth, respectively, among the world's most polluted countries in terms of the annual average $PM_{2.5}$ concentration ($\mu g/m^3$). Air pollution has become the third leading cause of death in Mongolia [13]. Frequent heavy smog incidents in Mongolia have attracted widespread public attention [14–17], and it has been reported that Mongolian children exposed to air pollution have poorer lung development and a higher prevalence of asthma [18,19]. Air pollution also negatively affects fetal growth, leading to low birth weight and preterm birth [20,21]. In UB, winter air pollution is also strongly associated with spontaneous abortion [22].

Over the past 30 years, there has been rapid growth of the urban population in Mongolia; the number of people living per square kilometer has increased 2.5-fold, from 117 in 1989 to 317 in 2019, and 46.1% of Mongolia's population (1.5 million people) lived in UB in 2019. Urbanization has brought enormous social and economic progress. It has improved infrastructure, health care, and educational resources, benefitting urban residents, but it has also brought problems, such as environmental pollution in the Ger Suburbs and urban areas [23–25]. For example, about 80% of UB's air pollution comes from about 3200 heating stoves in the Ger Suburbs [26]. Winter air pollution in UB has been a very serious problem for many years, with values many times higher than the WHO recommendations. For example, during the period from December 2016 to February 2017, the average concentration of $PM_{2.5}$ was 194 $\mu g/m^3$, and the maximum 24 h value reached 1065 $\mu g/m^3$ at the Bayankhoshuu site in the Ger Suburbs; these values are 3.9 and 7.8 times higher, respectively, than the Mongolian national air quality standard MNS 4585:2016 (50 $\mu g/m^3$) and WHO guidance level (25 $\mu g/m^3$).

The Mongolian government has made great efforts to solve the air pollution problem in UB. For example, MNT 164.1 billion and USD 104.7 million were invested in reducing air pollution between 2008 and 2016 [17]. On May 15, 2019, the Mongolian government implemented a ban on the burning of raw coal by UB households, and supplied "refined briquette" at a subsidized price close to that of raw coal; thus, the air quality is expected to improve.

Only a few reports on the ambient air quality of UB are available, and most of them were published before the implementation of the refined briquette program. These studies mainly analyzed just three pollutants ($PM_{2.5}$, $PM_{10}$, and $SO_2$), and there are almost no reports about the characteristics of the six pollutants $PM_{2.5}$, $PM_{10}$, $SO_2$, $NO_2$, CO, and $O_3$, the correlation between $PM_{2.5}$ and the other five pollutants, or the analysis of the types of air pollution in UB. In this study, through statistical analysis of automatic monitoring data for air pollutants in ambient air—namely, $PM_{2.5}$, $PM_{10}$, $SO_2$, $NO_2$, CO, and $O_3$—collected in UB from 2016 to 2019, we analyzed the characteristics of UB air pollutants and their influencing factors, while considering UB's natural environment, climate characteristics, energy structure, pollutant emission characteristics, and meteorological data. Our findings provide a theoretical basis for the prevention and control of air pollution in UB.

## 2. Materials and Methods

### 2.1. Overview of the Research Area

UB is the capital of Mongolia and the center of the country's development [24]. UB is located in the middle of the Mongolian Plateau at the southern end of the Kent Mountains on the banks of the Tula River—a tributary of the Orkhon River—at an altitude of 1351 m. It is known as the coldest capital in the world because of its geographical location. The

climate of UB is continental semi-arid, and is characterized by cold and long winters, and cool and short summers. The precipitation is highly variable and unevenly distributed. The annual average precipitation is 240–260 mm, and the summer precipitation from July to August accounts for about two-thirds of the annual precipitation [27]. UB mainly relies on coal combustion during an 8-month-long heating season (from 15 September to 15 May of the following year).

### 2.2. Materials and Methods

#### 2.2.1. Air Quality Index and Pollutant Concentration Limits

In October 2018, Mongolia's Ministry of Nature, Environment, and Tourism announced a new "Air Quality Index Air Quality Standard" (A1387). The air quality index (AQI) ranges and pollutant concentration limits are shown in Table 1.

**Table 1.** AQI ranges and pollutant concentration limits [28].

| AQI | Breakpoints | | | | | | | | | |
|---|---|---|---|---|---|---|---|---|---|---|
| | $SO_2$ $\mu g/m^3$ 24-h | $SO_2$ $\mu g/m^3$ 1-h | $NO_2$ $\mu g/m^3$ 24-h | $NO_2$ $\mu g/m^3$ 1-h | $PM_{10}$ $\mu g/m^3$ 24-h | $PM_{2.5}$ $\mu g/m^3$ 24-h | CO $mg/m^3$ 8-h | CO $mg/\mu m^3$ 1-h | $O_3$ $\mu g/m^3$ 8-h | $O_3$ $\mu g/m^3$ 1-h |
| 0–50 Good | - | 0–100 | - | 0–100 | 0–50 | 0–35 | 0–5 | - | 0–50 | - |
| 51–100 Moderate | - | 101–300 | - | 101–200 | 51–100 | 36–50 | 5–10 | 11–30 | 51–100 | - |
| 101–200 Unhealthy for Sensitive Groups | 50–800 | 301–800 | 50–280 | 201–700 | 101–300 | 51–150 | 11–15 | 31–60 | 101–265 | 250–400 |
| 201–300 Unhealthy | 801–1600 | 801–1600 | 281–565 | 701–2000 | 301–420 | 151–250 | 16–30 | 61–90 | 266–800 | 401–800 |
| 301–400 Very Unhealthy | 1601–2100 | 1601–2100 | 566–750 | 2001–3500 | 421–500 | 251–350 | 31–40 | 91–120 | - | 801–1000 |
| 401–500 Hazardous | 2101–2620 | 2101–2620 | 751–940 | 3501–3840 | 501–600, 601< | 351–500, 501< | 40–50 | 120–150 | - | 1001–1200 |

#### 2.2.2. Data Sources

The daily monitoring data for $PM_{2.5}$, $PM_{10}$, $SO_2$, $NO_2$, CO, and $O_3$ were obtained from of the Mongolian Ministry of Nature, Environment, and Tourism; the hourly monitoring data for $PM_{2.5}$ and $PM_{10}$ were obtained from the OpenAQ website (https://openaq.org/#/, accessed on 10 May 2022). The meteorological data came from the National Oceanic and Atmospheric Administration.

#### 2.2.3. Data Processing

The arithmetic mean of pollutant concentrations at all monitoring points in a city represents the overall mean value of pollutant concentrations in that city [29]. Some of the valid daily concentration data from 15 automatic air quality monitoring points in UB from 2016 to 2019 were lost to varying degrees. To more accurately show the real air quality of the city, we set missing data $\leq 25\%$ as a threshold for each monitoring point, and calculated the arithmetic average for the daily data that met this threshold, using it as the daily average concentration for each pollutant in UB. The hourly concentrations of $PM_{2.5}$ and $PM_{10}$ at the four automatic monitoring points in Tolgoit, Nisekh, Amgalan, and Bayankhoshuu were arithmetically averaged and used as the hourly concentration of PM in UB. Backward trajectory analysis was performed using the HYSPLIT model (http://ready.arl.noa.gov/HYSPLIT.php, accessed on 10 May 2022), and the correlation between $PM_{2.5}$ and the other five pollutants was calculated using SPSS statistical analysis software.

## 3. Results and Analysis

### 3.1. Ambient Air Quality

According to the UB AQI (Figure 1) data from 2016 to 2019, there was a total of 883 pollution days (Table 1); $PM_{2.5}$ was the primary pollutant for 553 of these days (60%), and $PM_{10}$ was the primary pollutant for 351 days (40%) (Table 2). Both spring and summer were dominated by $PM_{10}$ pollution, which was the primary pollutant on 75% and 91% of the total pollution days in spring and summer, respectively. In autumn, $PM_{2.5}$ and $PM_{10}$ were the primary pollutants for 49% and 51% of the total pollution days, respectively. In winter, $PM_{2.5}$ was the primary pollutant for 99% of the total pollution days. The number of days with hazardous and very unhealthy pollution days was 12 and 71, respectively. Hazardous pollution occurred on 1, 1, and 10 days in spring, autumn, and winter, respectively, while very unhealthy pollution occurred on 8 days in autumn and 63 days in winter.

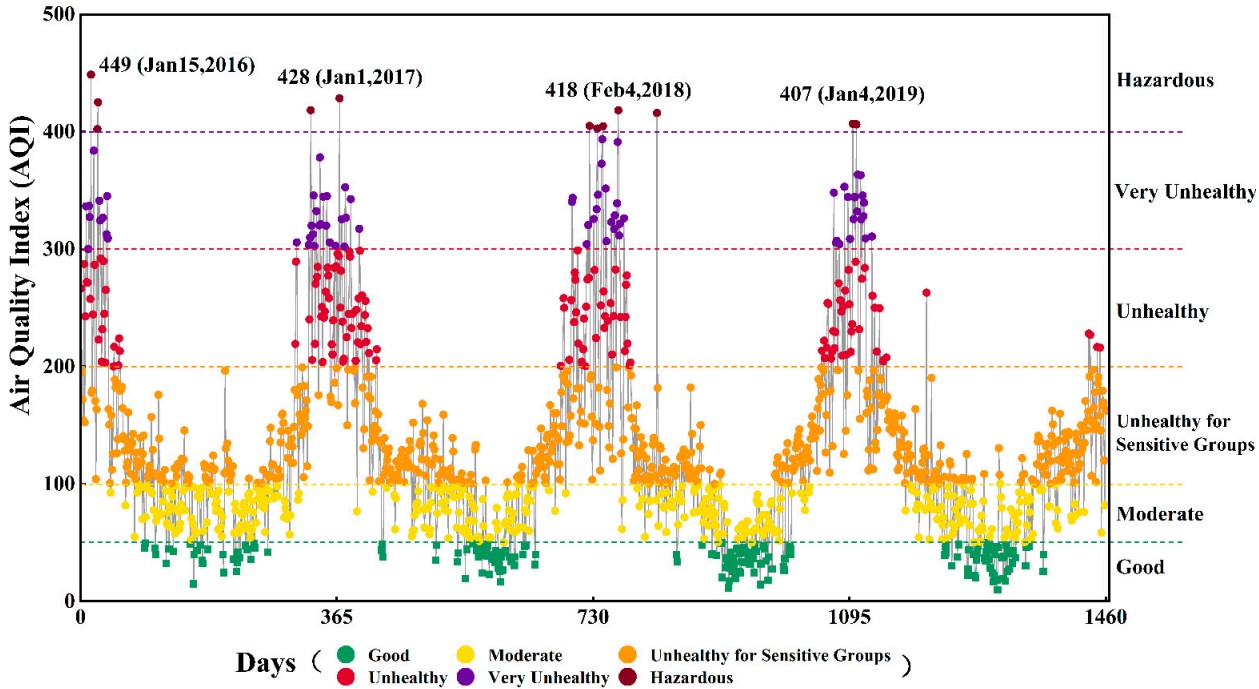

**Figure 1.** Air quality index in UB from 2016 to 2019.

**Table 2.** Monthly statistics and proportions of major pollutants during each season from 2016 to 2019.

| Primary Pollutant | Year | Spring | | | Summer | | | Autumn | | | Winter | | | Total |
|---|---|---|---|---|---|---|---|---|---|---|---|---|---|---|
| | | Mar. | Apr. | May | Jun. | Jul. | Aug. | Sep. | Oct. | Nov. | Dec. | Jan. | Feb. | |
| $PM_{2.5}$ | 2016 | 19 | - | - | - | 3 | - | 4 | 9 | 25 | 31 | 31 | 28 | 150 |
| | 2017 | 17 | - | - | - | 1 | - | - | 17 | 28 | 31 | 30 | 27 | 151 |
| | 2018 | 13 | - | - | - | - | - | - | 4 | 15 | 30 | 31 | 25 | 118 |
| | 2019 | 8 | 1 | - | - | 1 | 1 | - | 1 | 14 | 29 | 31 | 27 | 113 |
| | Total | 58 | | | 6 | | | 117 | | | 351 | | | 532 |
| | Proportion (%) | 25% | | | 9% | | | 49% | | | 99% | | | |
| $PM_{10}$ | 2016 | 5 | 14 | 10 | 8 | 11 | 5 | 6 | 10 | 3 | - | - | - | 72 |
| | 2017 | 7 | 22 | 16 | 12 | 5 | 1 | 7 | 8 | — | - | - | - | 78 |
| | 2018 | 10 | 17 | 21 | 6 | - | - | 8 | 19 | 13 | 1 | - | 1 | 96 |
| | 2019 | 20 | 16 | 12 | 8 | 1 | 1 | 15 | 21 | 10 | - | - | 1 | 105 |
| | Total | 170 | | | 58 | | | 120 | | | 3 | | | 351 |
| | Proportion (%) | 75% | | | 91% | | | 51% | | | 1% | | | |

In 2019, the annual average concentrations of $PM_{2.5}$, $PM_{10}$, $SO_2$, $NO_2$, $O_3$, and CO were 63.3 $\mu g/m^3$, 122.1 $\mu g/m^3$, 34.8 $\mu g/m^3$, 35.9 $\mu g/m^3$, 24.8 $\mu g/m^3$, and 1.41 $mg/m^3$, respectively. Compared with 2016, the average annual concentrations of $PM_{2.5}$, $PM_{10}$, $SO_2$, $NO_2$, and $O_3$ in 2019 decreased by 26%, 8.1%, 10%, 11.4%, and 25.5%, respectively, while the concentration of CO increased by 17.7% (Figure 2). The highest annual average concentrations of $O_3$ and CO were observed in 2017, and decreased thereafter, while the concentrations of the other four indicators decreased each successive year. During the four years investigated, the maximum daily average concentrations of $PM_{2.5}$, $PM_{10}$, $SO_2$, and $NO_2$ in UB were 8.5, 4.9, 4.2, and 2.2 times higher, respectively, than the 24 h standard limit of A1387, and the values of the remaining pollutants were never higher than the standard limit.

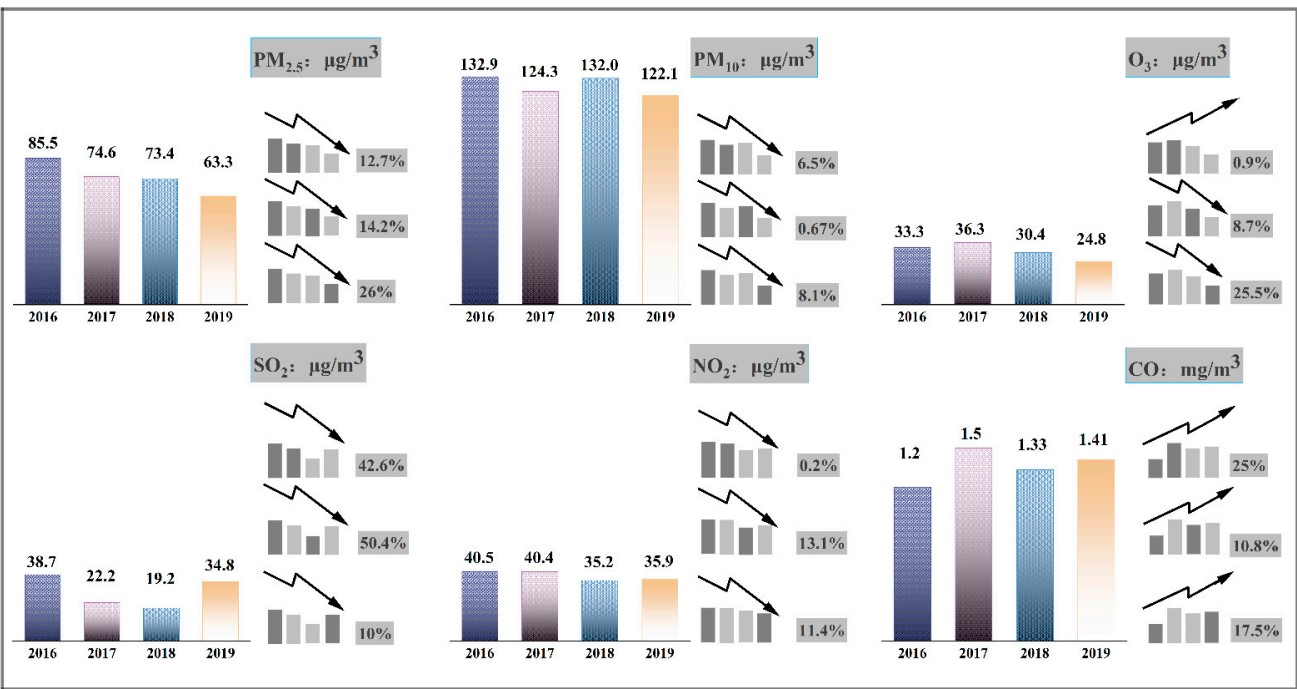

**Figure 2.** The interannual comparison of the average concentrations of six pollutants in UB from 2016 to 2019 (The annual average percentage change in each pollutant is calculated based on the concentration in 2016).

The percentages of days in 2016, 2017, 2018, and 2019 defined as having moderate or better air quality were 39.4%, 37.2%, 41.3%, and 40.2%, respectively. The proportion of days with good air quality increased from 7.7% in 2016 to 13.9% in 2019; the proportion of days with air quality that was unhealthy for sensitive groups increased from 40.3% to 50.7%, while the proportion of days with unhealthy or worse levels of pollution decreased from 20.5% to 9.0% (Figure 3). These data show that air quality improved year by year.

### 3.2. Variation in Particulate Matter over Time

$PM_{2.5}$ and $PM_{10}$ are produced by different sources. The $PM_{2.5}/PM_{10}$ ratio reveals the characteristics of particulate pollution, which can be used to characterize underlying atmospheric processes and assess historical $PM_{2.5}$ pollution without direct measurements [30]. For example, particulate pollution can be attributed to anthropogenic sources when $PM_{2.5}/PM_{10}$ values are high, while low $PM_{2.5}/PM_{10}$ ratios indicate substantial involvement of coarse particles, suggesting that the pollution is related to natural sources [31].

### 3.2.1. Year-to-Year Changes in Particulate Matter

PM concentrations fluctuated significantly from day to day, with average daily concentrations ranging from 3 to 423 $\mu g/m^3$ for $PM_{2.5}$, and from 10 to 516 $\mu g/m^3$ for $PM_{10}$

(Figure 4). During the four years of this study, the $PM_{2.5}$ concentration exceeded the 24 h A1387 limit (50 μg/m³) on 532 days, while the $PM_{10}$ concentration exceeded the limit (100 μg/m³) on 351 days (Table 2), indicating the severity of particulate pollution in UB. The obvious fluctuation in PM concentration caused the daily average $PM_{2.5}/PM_{10}$ ratio to vary greatly, between 0.12 and 1.14. The average ratios were 0.55, 0.52, 0.46, and 0.45 in 2016, 2017, 2018, and 2019, respectively, indicating that the proportion of $PM_{2.5}$ in particulate matter decreased year by year. This downward trend in particulate pollution occurred as the Mongolian government took a series of actions to reduce air pollution [17].

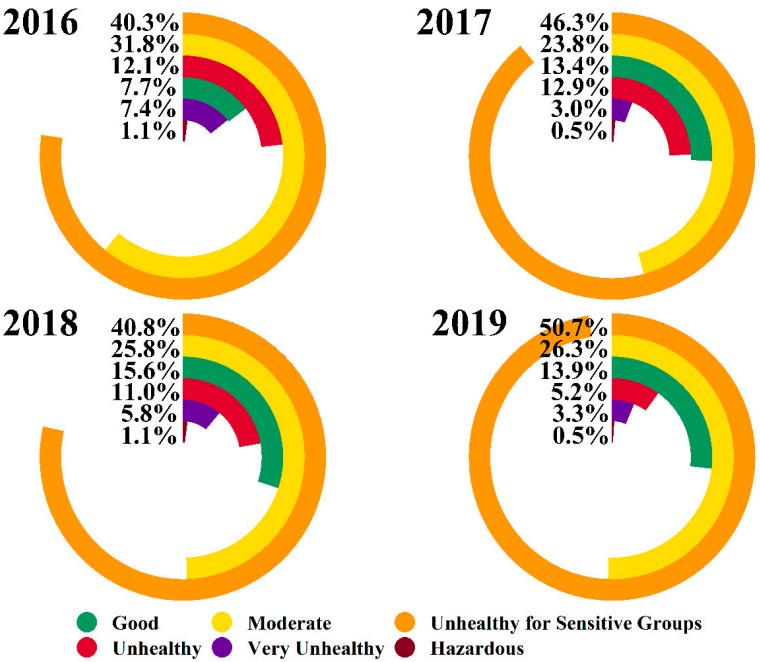

**Figure 3.** Percentages of days with different levels of air quality in UB from 2016 to 2019.

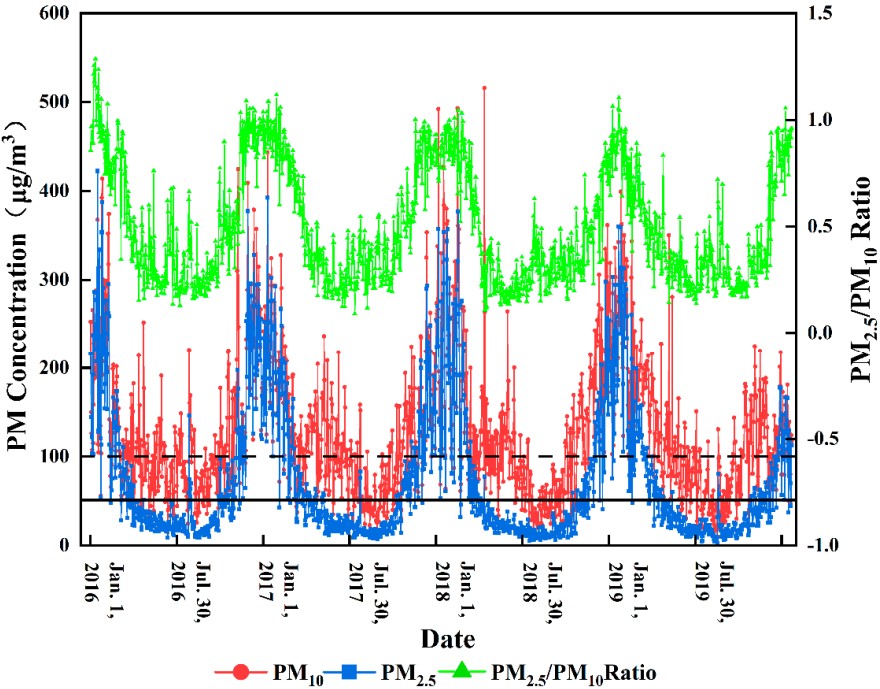

**Figure 4.** Daily average PM concentrations and $PM_{2.5}/PM_{10}$ ratios recorded from 2016 to 2019. The solid and dashed lines represent the 24 h limits for $PM_{2.5}$ and $PM_{10}$, respectively.

### 3.2.2. Seasonal Variation in Particulate Matter

Strong seasonal changes in the PM concentration and $PM_{2.5}/PM_{10}$ ratio were observed in UB. The $PM_{2.5}/PM_{10}$ ratio (Table 3) was highest in winter and lowest in summer in urban sites, and on days with moderate or better air quality in all four seasons. The PM concentration showed the following seasonal trend: winter > autumn > spring > summer. The average mass concentrations of $PM_{2.5}$ and $PM_{10}$ at urban sites in winter were 9.2 and 3.0 times higher than those in summer, and 3.3 and 1.7 times higher than those on days with moderate or better air quality, respectively. From 2016 to 2019, the PM concentrations and the $PM_{2.5}/PM_{10}$ ratios were relatively stable in spring and summer. However, compared with 2016, the average PM concentrations in autumn and winter in 2019 decreased, with $PM_{2.5}$ decreasing by 50.6% and 22.2%, $PM_{10}$ decreasing by 15.4% and 9.5%, and the $PM_{2.5}/PM_{10}$ ratio decreasing by 32.7% and 13.2%, respectively (Figure 5). The average value of $PM_{2.5}$ across all four seasons decreased by 89% from the highest level to the lowest, while the $PM_{10}$ concentration decreased by 67%. This suggests that there are stronger seasonal differences in $PM_{2.5}$ than in $PM_{10}$. The above seasonal changes may be due to differences in major pollution sources, emissions, and meteorological conditions in each season.

**Table 3.** Four-season average PM concentrations and $PM_{2.5}/PM_{10}$ ratios from 2016 to 2019 in UB.

| Period | Urban Sites (mean ± st. dev., μg/m³) | | | Good and Moderate Days (mean ± st. dev., μg/m³) | | |
|---|---|---|---|---|---|---|
| | $PM_{2.5}$ Conc. | $PM_{10}$ Conc. | $PM_{2.5}/PM_{10}$ | $PM_{2.5}$ Conc. | $PM_{10}$ Conc. | $PM_{2.5}/PM_{10}$ |
| Spring | 40 ± 17 | 114 ± 18 | 0.37 ± 0.14 | 20 ± 5 | 41 ± 3 | 0.49 ± 0.11 |
| Summer | 19 ± 4 | 68 ± 19 | 0.30 ± 0.05 | 13 ± 2 | 36 ± 3 | 0.36 ± 0.05 |
| Autumn | 63 ± 44 | 125 ± 40 | 0.47 ± 0.19 | 15 ± 3 | 38 ± 6 | 0.40 ± 0.10 |
| Winter | 175 ± 37 | 205 ± 34 | 0.85 ± 0.09 | 43 ± 6 | 61 ± 9 | 0.71 ± 0.14 |
| Full year | 74 ± 67 | 128 ± 57 | 0.50 ± 0.24 | 28 ± 21 | 49 ± 18 | 0.54 ± 0.24 |

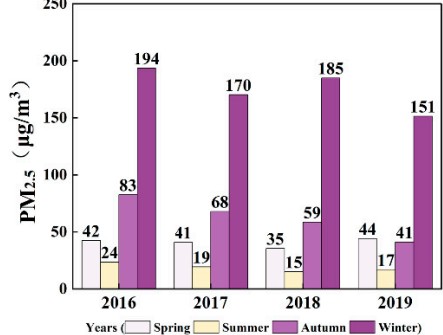 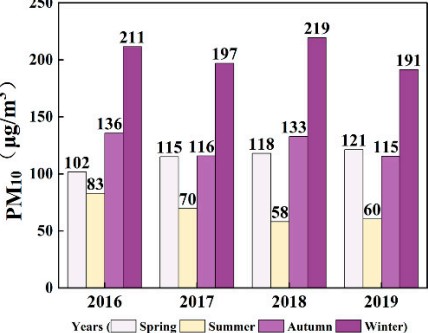 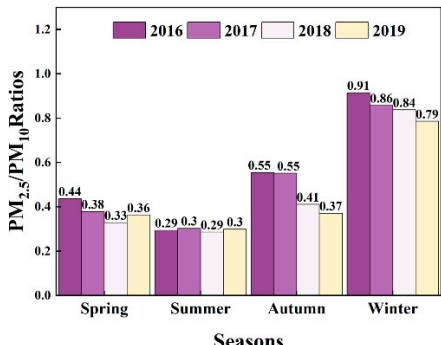

**Figure 5.** Seasonal changes in PM concentration and $PM_{2.5}/PM_{10}$ ratio.

An association between higher $PM_{2.5}/PM_{10}$ ratios and cooler seasons (autumn–winter) was previously found in a meta-analysis [32]. Increased domestic and industrial heating fuel consumption in winter leads to more fine particulate matter emissions [33]. The production of one of the main sources of fine particles—secondary aerosols—is accelerated due to the lower height of the mixed layer in winter [34]. Although the stable atmospheric conditions in winter are favorable for the dry deposition of coarse particles, they also increase the accumulation of fine particles in the air, leading to the dominance of $PM_{2.5}$ among the particles in winter [35].

### 3.2.3. Month-to-Month Changes in Particulate Matter

Both the $PM_{2.5}/PM_{10}$ ratio and PM concentration showed a "U"-shaped distribution (Figure 6). In all four years, the highest PM content was observed from December to January, while the lowest PM content was observed from June to August. In addition, the

PM content during the whole heating season was significantly higher than that during the non-heating season, indicating that coal-fired heating has a great impact on the ambient air quality in UB. The PM concentration was lowest from June to August, which may be related to the cessation of coal-fired heating and the increase in precipitation [36,37]. From March to September, during which $PM_{10}$ is greatly affected by high-speed winds and frequent sandstorms, the average $PM_{2.5}/PM_{10}$ ratio was only 0.33, which is significantly lower than that in other months. Coarse particulate matter pollution was clearly observed in spring and summer.

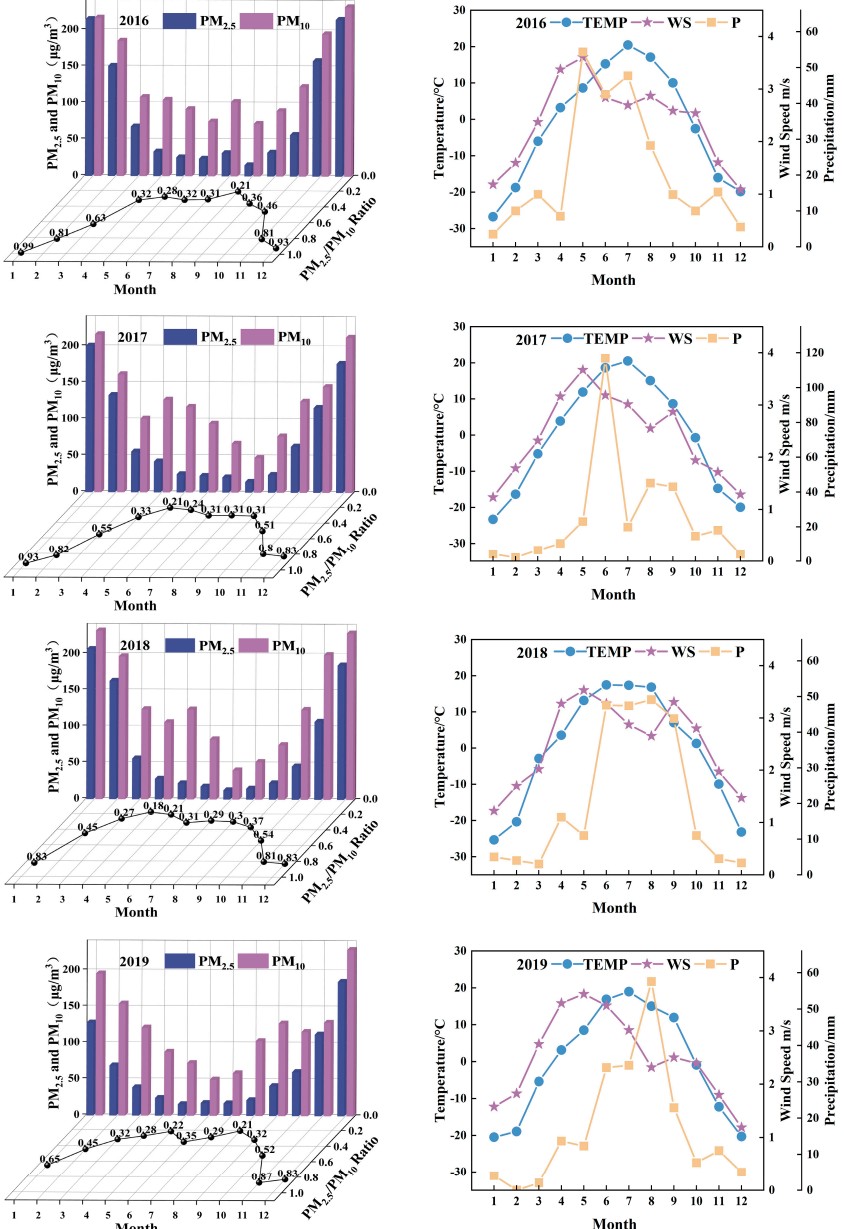

**Figure 6.** Monthly changes in pollutants and meteorological parameters (TEMP: temperature, WS: wind speed, P: precipitation).

### 3.2.4. Hourly Variation in Particulate Matter

There was a clear diurnal difference in the $PM_{2.5}/PM_{10}$ ratio (Figure 7), which increased from 16:00 to 04:00, with a peak of 0.79, and then decreased until 15:00 during the day. Therefore, temperature changes should be considered in order to better understand the apparent diurnal differences in $PM_{2.5}/PM_{10}$ ratios. During the night, stable atmospheric conditions caused by temperature inversions restrict vertical airflow, and promote the

dry deposition of coarse particles and the accumulation of fine particles [38]; thus, the $PM_{2.5}/PM_{10}$ ratio gradually increases at night. During daytime, the $PM_{2.5}/PM_{10}$ ratio gradually decreases due to resuspension of coarse road dust and human activities.

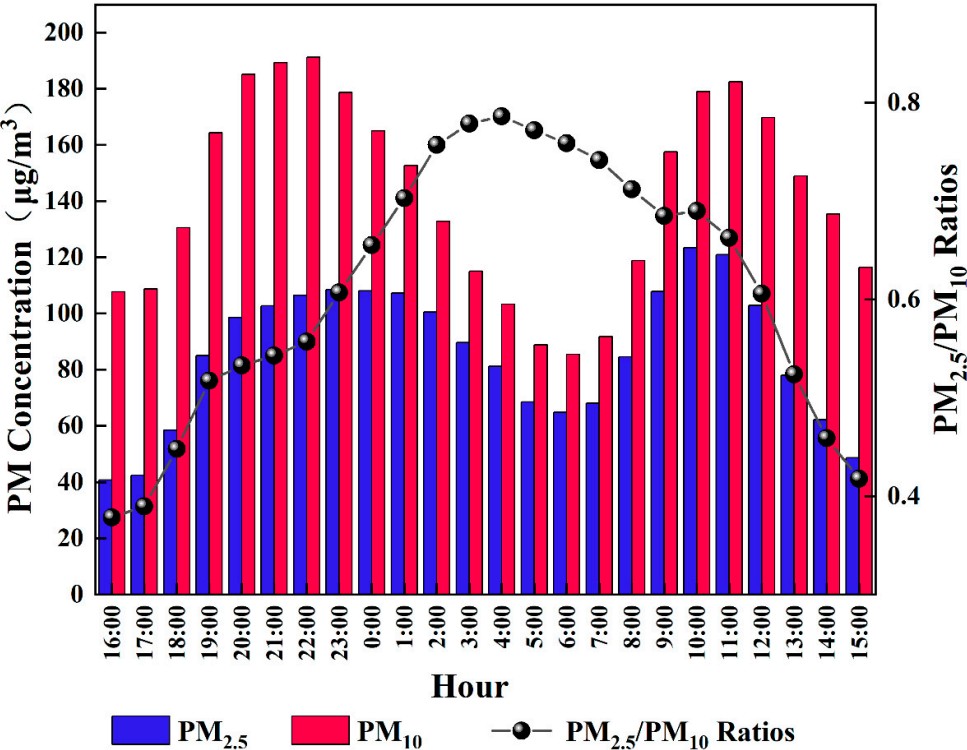

**Figure 7.** Daily changes in the $PM_{2.5}/PM_{10}$ ratio and PM concentration at urban sites from September 2017 to February 2019.

The distribution of daily variation in PM concentration was bimodal. The first peaks of $PM_{2.5}$ and $PM_{10}$ occurred at 10:00 and 11:00, respectively, and these peaks were associated with increased PM concentrations due to cooking, heating, traffic recovery, and particulate emissions. The second peaks occurred at 23:00 and 22:00, respectively, and were possibly due to nighttime emissions [17,39].

The bimodal pattern of PM concentration in UB is very similar to that observed for other cities, such as Beijing–Tianjin–Hebei in China [40] and Seoul in South Korea [41]. The increase in the height of the boundary layer and the decrease in the thickness of the inversion layer during the day make diffusion of the pollutants easier [16], so the concentration of ground pollutants decreases in the afternoon.

To be consistent with the recording times in Figure 7, we also used 16:00 as the starting time for calculating the $PM_{2.5}/PM_{10}$ ratio for visualization of seasonal changes (Figure 8). The diurnal trends for the four seasons were relatively similar; the average values of $PM_{2.5}/PM_{10}$ in spring, summer, fall, and winter were 0.32, 0.29, 0.57, and 0.88, respectively. By comparison, a previous study found that the increase in $PM_{2.5}$ concentration in winter directly led to an increase in the $PM_{2.5}/PM_{10}$ ratio, which also confirmed the relationship between secondary particulate matter and $PM_{2.5}/PM_{10}$ [42]. Meanwhile, other studies have demonstrated that $PM_{2.5}/PM_{10}$ reflects the degree of enrichment of fine particles; the larger the ratio, the more serious the levels of secondary pollutants in the city [43,44].

The distribution of hourly PM concentration in UB exhibited a bimodal pattern in all four seasons, with a peak appearing after the commuting rush hour, indicating that vehicle exhaust emissions and human activities have an obvious impact on PM pollution. Moreover, the PM concentration during the entire heating season was significantly higher than that during the non-heating season, indicating that coal-fired heating has a great

impact on the ambient air quality. The current urban energy structure—dominated by coal burning—is the main reason for this phenomenon [45].

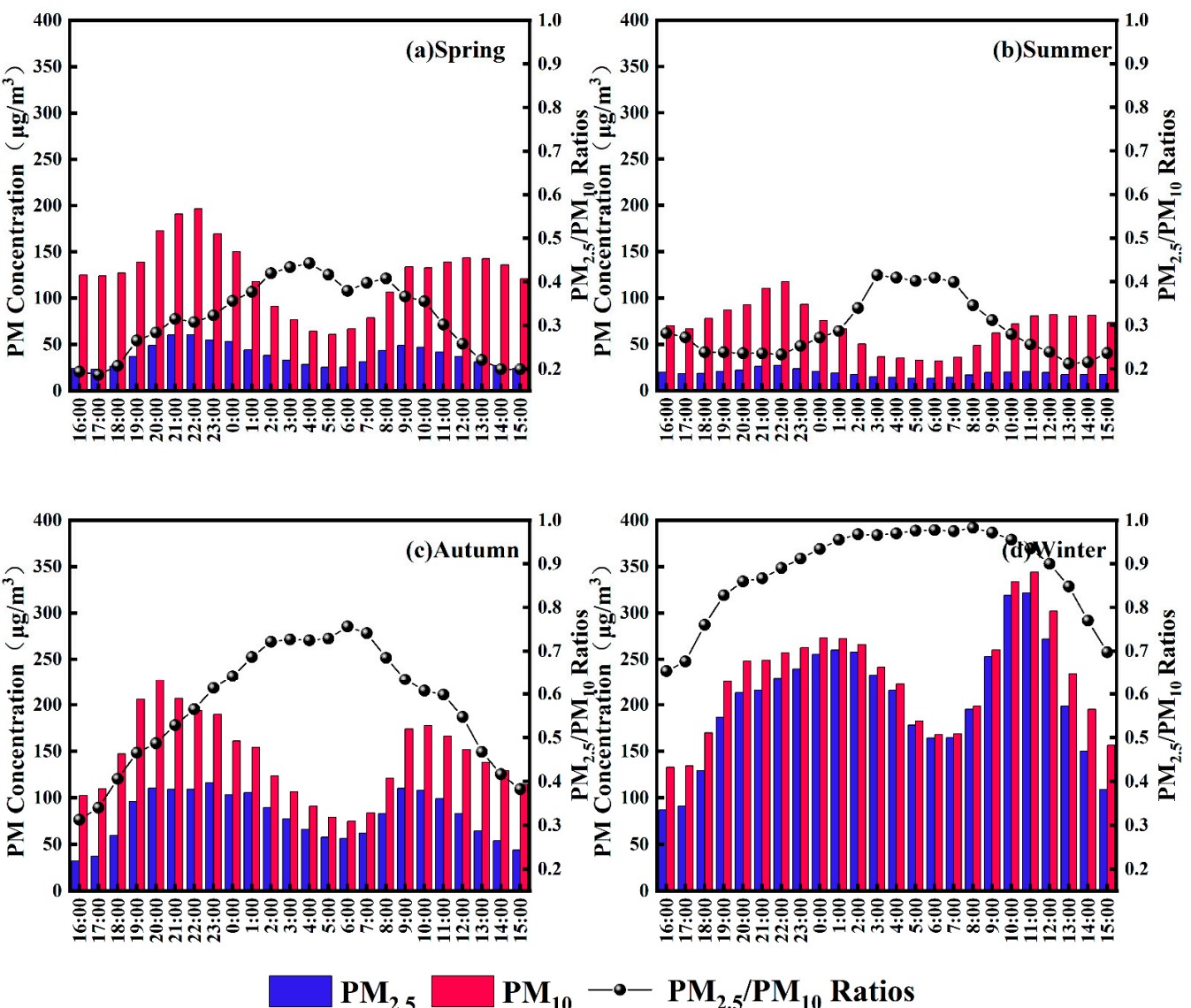

**Figure 8.** Hourly changes in the $PM_{2.5}/PM_{10}$ ratio and PM concentration at urban sites in all four seasons from September 2017 to February 2019 (**a–d**).

### 3.3. Analysis of Pollution Types

From 2016 to 2019, 12 hazardous pollution days occurred in UB, 83% of which were in winter. A 48 h backward trajectory analysis of the process of hazardous pollution in UB from 2016 to 2019 was carried out using NOAA HYSPLIT. In the past four years, the movement of severely polluted weather clusters in different seasons in UB has changed significantly. The direction of the air masses is mainly west in spring, while it is northwest and due north in autumn and winter, and there are frequent wind changes near the ground (Figure 9). The data demonstrated that the air mass in the north further aggravated the degradation of air quality in autumn and winter (Table 4). On 31 March 2018, the average $PM_{2.5}$ and $PM_{10}$ values in UB were 78 $\mu g/m^3$ and 516 $\mu g/m^3$, respectively, while the $PM_{2.5}/PM_{10}$ ratio was only 0.15, indicating that coarse particle pollution caused by sand and dust was high, but the concentrations of other pollutants were low. The average values of $PM_{2.5}$ and $PM_{10}$ on heavily polluted days in autumn and winter reached as high as 372 $\mu g/m^3$ and 401 $\mu g/m^3$, respectively, while the $PM_{2.5}/PM_{10}$ ratio was 0.94. The

hazardous pollution days were mainly caused by a combination of high-intensity emissions from coal-combustion-induced sources and unfavorable meteorological conditions. As reported previously [16,46], both the thickness and intensity of the inversion layer reached their maximum values (exceeding 500 m) in January in UB, and showed seasonal variation similar to that of the PM concentration. At the same time, the monthly average temperature inversion intensity had a strong positive correlation with the monthly average PM$_{2.5}$ concentration. The enhanced radiative cooling of UB's basin-like terrain led to a stable atmosphere in urban areas, which further aggravated particulate air pollution. To sum up, sand dust and soot are the two main types of hazardous pollution in UB.

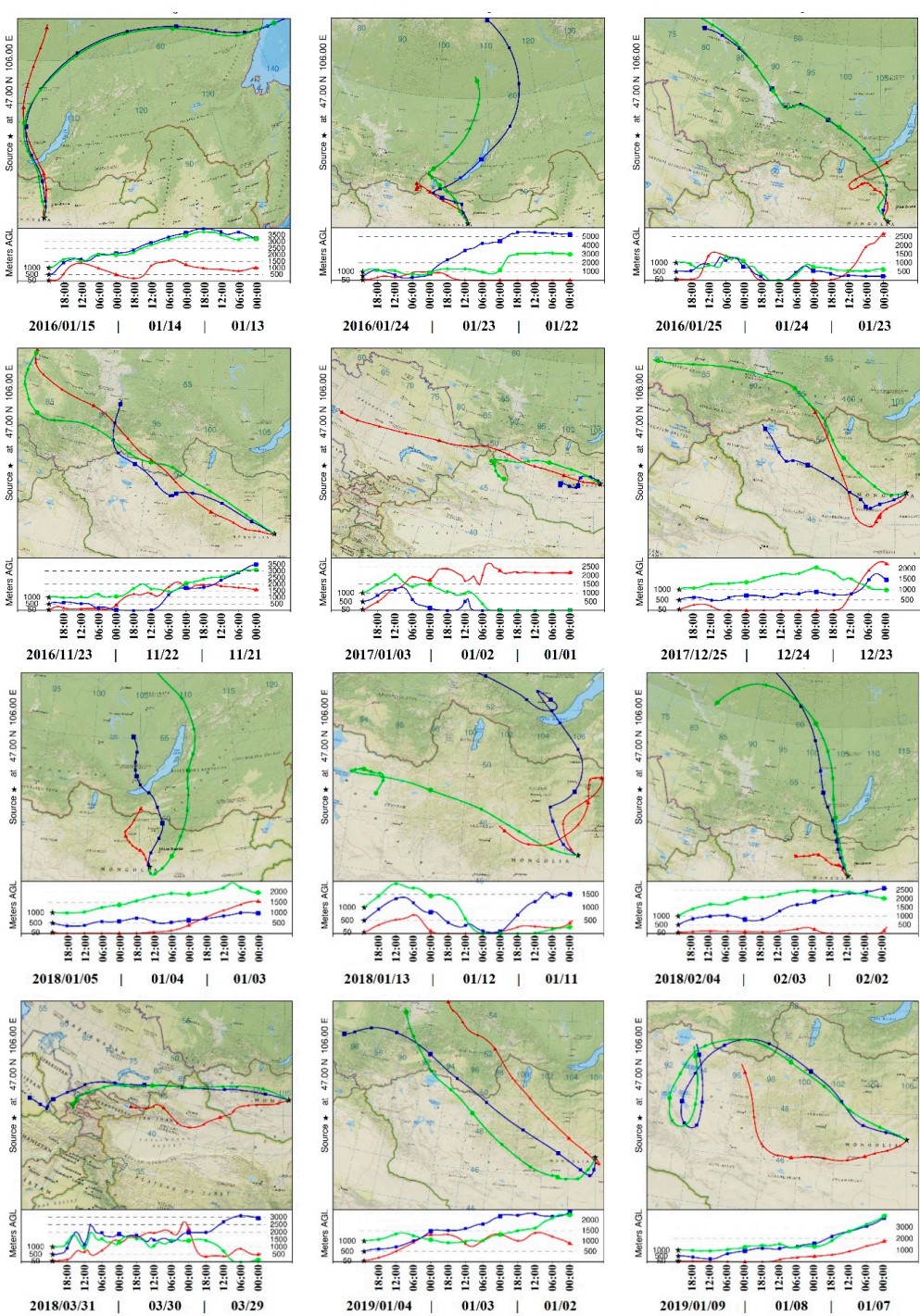

**Figure 9.** A 48 h backward trajectory analysis of heavily polluted days from 2016 to 2019.

**Table 4.** Timeline of the heavy pollution process.

| Season | Date | AQI | Prime Pollution | $PM_{10}$ | $PM_{2.5}$ | $SO_2$ | $NO_2$ | CO | $O_3$ | $PM_{2.5}/PM_{10}$ | Direction of the Wind |
|---|---|---|---|---|---|---|---|---|---|---|---|
| Spring | 31 March 2018 | 416 | $PM_{10}$ | 516 | 78 | 15 | 39 | 0.7 | 42 | 0.15 | West |
| Summer | / | / | / | / | / | / | / | / | / | / | / |
| Autumn | 23 November 2016 | 418 | $PM_{2.5}$ | 409 | 377 | 97 | 84 | 4.3 | 7 | 0.92 | Northwest |
| Winter | 15 January 2016 | 449 | $PM_{2.5}$ | 368 | 423 | 161 | 81 | 4.3 | 5 | 1.15 | Shift from north to east |
| | 24 January 2016 | 402 | $PM_{2.5}$ | 392 | 353 | 165 | 70 | 3.8 | 7 | 0.9 | Shift from west to north |
| | 25 January 2016 | 425 | $PM_{2.5}$ | 414 | 387 | 156 | 72 | 3.8 | 6 | 0.94 | Northwest (frequent wind shifts near the ground) |
| | 03 January 2017 | 428 | $PM_{2.5}$ | 443 | 393 | 121 | 98 | 8.6 | 6 | 0.89 | Northwest |
| | 25 December 2017 | 405 | $PM_{2.5}$ | 492 | 357 | 69 | 81 | 5.7 | 9 | 0.73 | Shift from southwest to north |
| | 05 January 2018 | 403 | $PM_{2.5}$ | 427 | 354 | 74 | 65 | 3.8 | 10 | 0.83 | North |
| | 13 January 2018 | 405 | $PM_{2.5}$ | 366 | 357 | 70 | 67 | 4.4 | 10 | 0.98 | Northwest (frequent wind shifts near the ground) |
| | 04 February 2018 | 418 | $PM_{2.5}$ | 361 | 377 | 55 | 63 | 3.5 | 13 | 1.04 | North |
| | 04 January 2019 | 407 | $PM_{2.5}$ | 343 | 360 | 71 | 61 | 4.0 | 7 | 1.05 | Northwest |
| | 09 January 2019 | 406 | $PM_{2.5}$ | 399 | 359 | 100 | 81 | 5.1 | 26 | 0.90 | Shift from west to north (frequent wind shift at high altitude) |

*3.4. The Relationship between $PM_{2.5}$ and Five Other Pollutants*

Spearman's correlation test was used to determine the relationship between $PM_{2.5}$ and five other pollutants. In this study, the level of correlation was determined by referring to the correlation coefficient value. A value between 0.0 and 0.25 was considered as low correlation, 0.26–0.50 as fair correlation, 0.51 to 0.75 as moderate correlation, and 0.75–1.00 as high correlation [47]. The Spearman's correlation coefficients between daily average $PM_{2.5}$ and $PM_{10}$, $SO_2$, $NO_2$, and CO were 0.851, 0.855, 0.861, and 0.871, respectively, indicating a significant positive correlation between pollutants. There was a negative correlation between $PM_{2.5}$ and $O_3$, with a correlation coefficient of $-0.646$. The daily and monthly mean concentrations of $PM_{2.5}$ were fitted against those of $PM_{10}$ (Figure 10), and the $R_2$ values were 0.763 and 0.942, respectively, indicating a strong linear correlation between $PM_{2.5}$ and $PM_{10}$.

There were obvious seasonal differences in the Spearman's correlation coefficients between $PM_{2.5}$ and $PM_{10}$ (Table 5), which were closely related to external factors such as meteorological conditions and manmade pollution. The correlation coefficients in spring, summer, autumn, and winter were 0.502, 0.751, 0.883, and 0.938, respectively. Spring (Figure 11a) was the season with the lowest linear correlation, with an $R^2$ value of only 0.187. The distribution of points on both sides of the fitted line in spring was relatively uneven, indicating that some sources of $PM_{2.5}$ and $PM_{10}$ pollution were different in spring, and the concentration of $PM_{10}$—which is mainly related to sources of dust—was relatively high. The $PM_{2.5}$ and $PM_{10}$ data points from summer (Figure 11b) were concentrated, and the 4-year averages were $19 \pm 4$ and $68 \pm 19$, respectively (Table 3), indicating that the pollution sources were relatively fixed in summer. In addition, UB has abundant precipitation and high wind speeds in summer, which help to quickly dilute and diffuse pollutants, and contribute to the low levels of pollution. The data points for autumn (Figure 11c) were scattered and distributed in both medium and high concentrations. Initial heating and biomass burning were the main reasons for the increase in the average value

throughout the autumn. The main reason for the higher PM concentrations in winter (Figure 11d) was the significant increase in coal burning for heating, resulting in increased emissions of particulate matter and its precursors. In addition, the formation of a stable atmosphere in winter allowed pollutants to accumulate, and created pollution events.

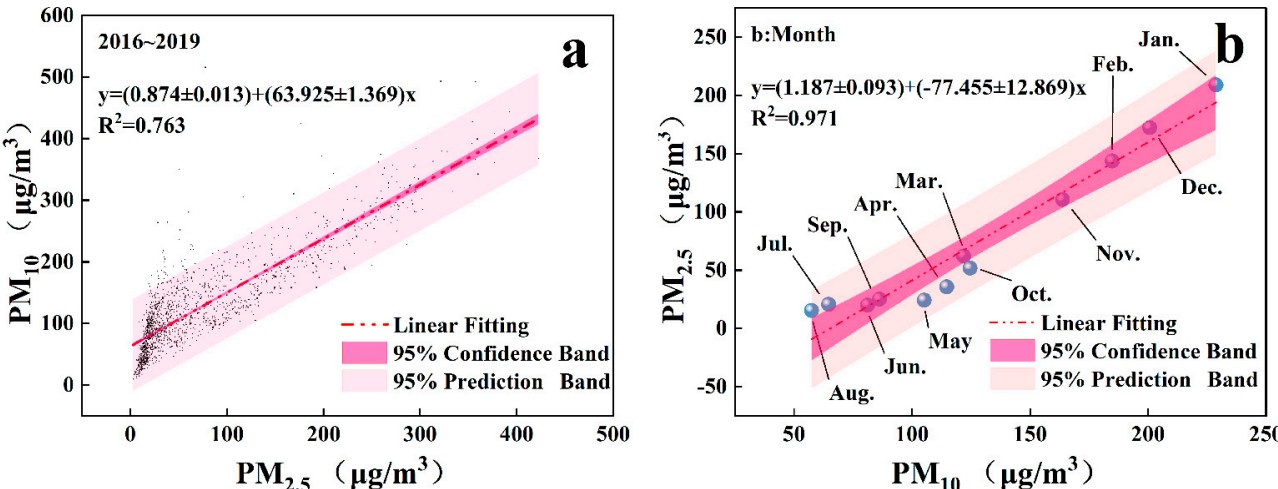

**Figure 10.** The linear relationship between $PM_{2.5}$ and $PM_{10}$ in terms of daily (**a**) and monthly (**b**) mean concentrations.

**Table 5.** Spearman's correlation coefficients between $PM_{2.5}$ and five other pollutants in different seasons.

| Season | $PM_{10}$ | $SO_2$ | $NO_2$ | CO | $O_3$ |
|---|---|---|---|---|---|
| Spring | 0.502 | 0.854 | 0.667 | 0.667 | −0.400 |
| Summer | 0.751 | 0.370 | 0.431 | 0.161 | 0.310 |
| Autumn | 0.883 | 0.883 | 0.909 | 0.921 | −0.800 |
| Winter | 0.938 | 0.403 | 0.630 | 0.726 | −0.559 |

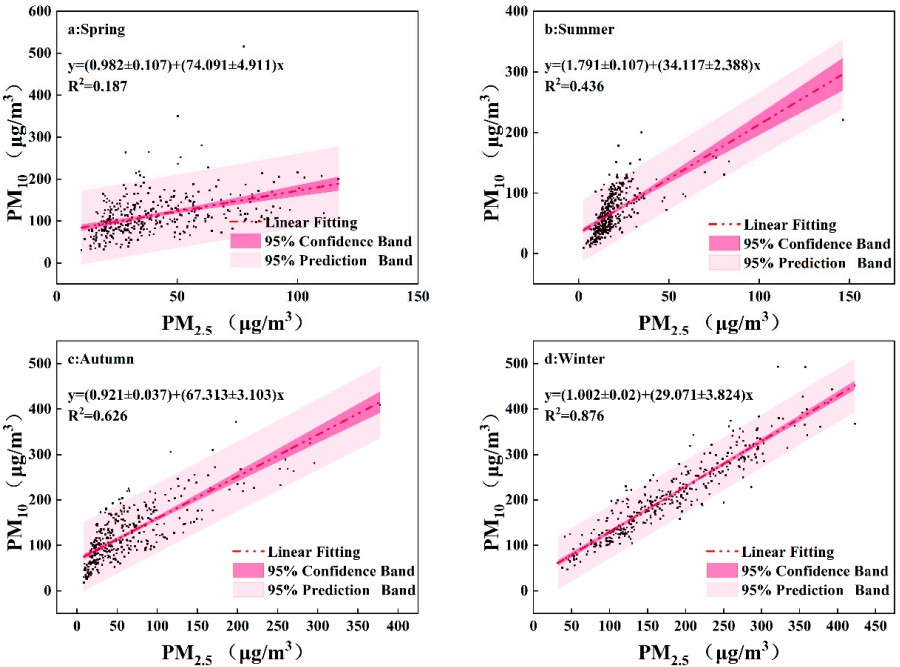

**Figure 11.** Quasi-linear relationships between $PM_{2.5}$ and $PM_{10}$ in four different seasons. (**a**): Spring; (**b**): Summer; (**c**): Autumn; (**d**): Winter.

PM$_{2.5}$ was significantly positively correlated with SO$_2$, NO$_2$, and CO in all four seasons, to varying degrees; however, it was positively correlated with O$_3$ in summer but negatively correlated with O$_3$ in winter. PM$_{2.5}$ in the atmosphere arises not only from direct emissions from pollution sources, but also from secondary pollutants such as sulfates, nitrates, and organic aerosols, which are produced by the homogeneous or heterogeneous (at the particle surface) reaction of gaseous precursors such as SO$_2$ and NO$_x$ in the atmosphere [48].

The positive correlation of PM$_{2.5}$ with SO$_2$, NO$_2$, and CO indicated that SO$_2$ and NO$_2$ generate sulfates and nitrates through homogeneous or heterogeneous reactions which, in turn, have an important impact on the mass concentration of PM$_{2.5}$. The aerosol extinction formed by PM$_{2.5}$ in winter inhibits the generation of O$_3$, so PM$_{2.5}$ and O$_3$ are negatively correlated in winter (i.e., the opposite to what is observed in summer). In addition, PM$_{2.5}$ has the same source as the three aforementioned gaseous pollutants (Table 5). SO$_2$ mainly comes from the combustion of fossil fuels, NO$_2$ mainly comes from vehicle emissions and coal combustion, and CO mainly comes from the metallurgical industry, internal combustion engine exhausts, and incomplete combustion of fossil fuels. Therefore, coal combustion and vehicle exhaust emissions are important factors influencing PM$_{2.5}$ in UB.

## 4. Conclusions and Future Perspectives

Accurate assessment of individual ambient air pollution exposure levels is a key part of epidemiological research aimed at studying the adverse health effects of poor air quality. This is especially challenging in developing countries with heavy air pollution, mainly because of sparse monitoring networks and a lack of consistent data.

In this study, we analyzed the air quality, temporal variation in particulate matter, and the correlation between pollution type and pollutants in UB from 2016 to 2019. Obvious seasonal and diurnal differences in PM concentrations were observed. The distributions of hourly PM concentrations showed a bimodal pattern over the course of each year, with the highest concentration observed in winter. In the four years of the study, 83% of the severe pollution days occurred in winter. In addition to high-intensity emissions during the heating season, energy structure, vehicle exhausts, topography, and meteorology are also important factors that further aggravate air pollution in UB.

Air pollution is a global problem that we must all tackle together, and reducing pollution from the original source can produce quick and substantial effects. In the winter after the Mongolian government implemented the ban on household consumption of raw coal in UB in May 2019, the number of days with unhealthy or worse pollution decreased significantly compared with the numbers in previous years. Air quality improved in UB in 2019 compared with that in 2016. Specifically, the number of days with good air quality increased by 45%; the number of days with unhealthy or worse levels of pollution decreased by 56%; the annual average PM$_{2.5}$, PM$_{10}$, SO$_2$, and NO$_2$ concentrations decreased by 26%, 8.1%, 10%, and 11.4%, respectively; and the average concentrations of PM$_{2.5}$, PM$_{10}$, SO$_2$, and NO$_2$ during the heating season decreased by 25.7%, 4.6%, 11.2%, and 11%, respectively.

From the references that we have gained access to so far, only a select few report using models to predict PM$_{2.5}$ concentrations in UB. Now that we have a full understanding of the air quality situation in UB and its influencing factors, we are currently using MATLAB software to combine the automatic ambient air quality monitoring data with meteorological and aerosol data to establish a PM$_{2.5}$ prediction model to provide data support for the improvement of air quality in UB.

**Author Contributions:** Conceptualization, S. (Suriya), A. and H.Z.; methodology, S. (Suriya); software, S. (Suriya) and S. (Sachurila); validation, S. (Suriya), N.N. and H.Z.; formal analysis, S. (Suriya); investigation, S. (Suriya); resources, S. (Suriya); data curation, S. (Suriya); writing—original draft preparation, S. (Suriya); writing—review and editing, S. (Suriya); visualization, S. (Suriya); supervision, S. (Suriya); project administration, S. (Suriya); funding acquisition, S. (Suriya). All authors have read and agreed to the published version of the manuscript.

**Funding:** This research received no external funding.

**Institutional Review Board Statement:** Not applicable.

**Informed Consent Statement:** Not applicable.

**Data Availability Statement:** No new data were created or analyzed in this study. Data sharing is not applicable to this article.

**Conflicts of Interest:** The authors declare no conflict of interest.

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
