# Peer review of "Spatiotemporal Variation in Air Pollution Characteristics and Influencing Factors in Ulaanbaatar from 2016 to 2019"

_atmosphere, doi:10.3390/atmos13060990_

Round 1
Author Response
Dear Reviewer:
Thank you very much for your letter dated 25 May 2022, and the referees’ reports. The co-authors and I would like to thank you for the time and effort spent in reviewing the manuscript. Based on comments from you and the reviewers, we have made extensive modifications to the original manuscript. Each comment will be directly addressed regarding the modified manuscript with changes highlighted in yellow. Here, we attach the revised manuscript in the format of MS Word for your approval. A document answering every question from the referees is also summarized and enclosed. Should you have any questions, please contact us without hesitation.
Reply to the comments on atmosphere-1745168:
- The title seems weird, it does not communicate the real idea of the work.
Reply: Thank you for your correction, we changed the title to Spatio-temporal variation in air pollution characteristics and influencing factors in Ulaanbaatar from 2016 to 2019.
- The abstract seems ok but please add one more introductory line of your objective at beginning of the abstract.
Reply: Thank you for your suggestions, we have followed it and added one more introductory line of your objective at beginning of the abstract.
- Keywords are wrongly formatted. Please change according to journal guidelines.
Reply: We have made changes in the article .
- Research gap should be delivered in a more clear way with the directed necessity for future research work.
Reply: Thanks for this suggestion. We have modified the text in the manuscript to make it clear about necessity for future research work.
- Introduction section must be written in a more quality way, i.e., more up-to-date references addressed. Please target the specific gap such as 2015-2021 etc.
Reply: Sorry about this. We have updated the references we cited in the introduction as you suggested.
- The novelty of the work must be clearly addressed and discussed, compare previous research with existing research findings and highlight novelty.
Reply: Appreciate your valuable suggestion. We have further clarified the novelty of our study and how it differs from previous studies at the end of the introductionn.
- Please check the abbreviations of words throughout the article. All should be consistent.
Reply: Thank you for your correction, we have made changes in the article.
- Please don’t use lumpy references (such as: [10-12]). Each reference needs to be properly addressed. Please revise your paper accordingly since the same issue occurs in several spots in the paper.
Reply: Thank you for your correction, we have made changes in the article.
- The main objective of the work must be written in the more clear and more concise way at the end of the introduction section.
Reply: Thank you for your correction, we have made changes in the article.
- Section 1.1. in the experimental art has different dates without any reference. Please provide accurate sources of the provided information. Please revise your paper accordingly since some issue occurs in several spots in the paper.
Reply: In the section 1.1 we have added reference.
- Please provide space between numbers and units. Please revise your paper accordingly since some issue occurs in several spots in the paper.
Reply: Thank you for your correction, we have made changes in the article.
- Regarding the replications, the authors confirmed that replications of the experiment were carried out. However, these results are not shown in the manuscript, how many replicates were carried out by experiment? Results seem to be related to a unique experiment. Please, clarify whether the results of this document are from a single experiment or from an average resulting from replications. If replicated were carried out, the use of average data is required as well as the standard deviation in the results and figures shown throughout the manuscript. In the case of showing only one replicate explain why only one is shown and include the standard deviations.
Reply: The data in Table 3 are the average PM concentrations and PM2.5/PM10 ratios at urban sites. The concentrations and ratios are the averages from four days in each of the four seasons in each year. The means and standard deviations are shown in the table. The purpose of these data is to verify that PM concentrations and ratios show the same strong seasonal changes at urban sites and in good seasons.
- Please provide a high-quality image for figures 1,2,3, 9, 10 and 11. Also, figure 2 and 9 are not well explained please highlight the main points.
Reply: We have updated figures 1, 2, 3, 9, 10, and 11 with high-quality images. And we also made further explanations for figures 2 and 9.
- Please add a comparative profile section to compare your results and prove how it better than the previous.
Reply: We have made changes in the article.
- Section 3 should be renamed by Conclusion and Future perspectives. The conclusion section is missing some perspective related to the future research work, quantifying the main research findings, and highlighting the relevance of the work with respect to the field aspect. In present form chapter, 3 is totally weird. This chapter 3 should be chapter 4.
Reply: The conclusion part of Chapter 3 in the original manuscript did contain a summary and an outlook for future work. We have changed it to “Conclusion and future perspectives” according to your suggestion.
- To avoid grammar and linguistic mistakes, Major level English language should be thoroughly checked. Please revise your paper accordingly since several language issues occurs on several spots in the paper.
Reply: Thank you for your correction, we have made changes in the article.
- Reference formatting needs carefully revision. All must be consistent in one formate. Please follow the journal guidelines. Overall, the author should carefully rewrite the work and format it again in accordance with the journal. There are several formatting errors throughout the manuscript.
Reply: Thank you for your correction, we have made changes in the article.
All the lines and pages indicated above are in the revised manuscript.
Thank you and all the reviewers for the kind advice.
Sincerely,
Suriya
Department of Chemistry School of Mathematics and Natural Science Mongolian National University of Education
Ulaanbaatar 210648, Mongolia
Telephone: +976-95819724
Email: suriyaa@ hlbec.edu.cn

Reviewer 2 Report
Suriya et al. analyzed years of air monitoring data taken in Ulaanbaatar from 2016 and 2019 and examined the close relationship between PM2.5 and other pollutants. Overall the authors have done a great job explaining the methodology and approach, and presented the results in a very clear manner. There are a few minor concerns that could be addressed before the paper can be considered for publication.
Table 1: Please consider removing the background color using lighter colors.
Figure 2: It was a bit unclear how the percentage was calculated. I would assume everything was calculated using 2016 as the base year, but please consider adding a note in the caption.
Figure 5: there seems to be no reason to use 3D graphing in the first two figures. It doesn’t help with visualization.
Figure 6: WS as wind speed and P as precipitation need to be properly defined before using on figures directly.
Figure 10: Daily correlation between PM2.5 and PM10 seems to follow a different/more steep linear trend when PM2.5 is less than 50 ug/m3. It might be helpful for the authors to separate the two trends in case there is a threshold for PM10/PM2.5 ratio to shift. Same with summer and autumn trends shown in Figure 11.
Reviewer 3 Report
In the manuscript the authors presented the air pollutant data obtained from automatic measuring station in Ulaanbaatar city from 2016-2019. Temporal and seasonal variations of PM10, PM2.5, SO2, NO2, CO, and O3 together with inter-annual comparison were analyzed. The correlation between pollutants is also presented. In addition, hourly variation in PM was described and for several heavy polluted days a 48-hour backward trajectory were presented indicating possible transport contribution.
The obtained results shows that the air quality in UB improved significantly in 2019 compared with that in 2016 as a result of different mitigation applied.
This is a very good written manuscript, but from my point of view rather technical. The authors use routine available measurements and descriptive statistics to show basic variations, correlations and trends. There are no new experimental/theoretical approach, but standard very good written report. Although it can be useful for local/regional air quality strategy, there is no results that could be of wider scientific interest (there are many data from the cities worldwide analyzed in the similar way through standard air quality plans/reports).
I suggest, based on the obtained results, to further analyze the database but using more robust techniques like AI or machine learning algorithm to investigate in depth the relation between pollutants and other parameters (meteorology…).
Some minor technical corrections:
Figure 3. Percentage of days with different levels of air quality in UB from 2016 to 2019.
There is no Figure 3 in the manuscript. Please check it.
Line 178: “…and PM2.5/PM10 …” Please use subscripts
Line 196: same as previous suggestion
Author Response
Dear Reviewer:
Thank you very much for your letter dated 25 May 2022, and the referees’ reports. The co-authors and I would like to thank you for the time and effort spent in reviewing the manuscript. Based on comments from you and the reviewers, we have made extensive modifications to the original manuscript. Each comment will be directly addressed regarding the modified manuscript with changes highlighted in yellow. Here, we attach the revised manuscript in the format of MS Word for your approval. A document answering every question from the referees is also summarized and enclosed. Should you have any questions, please contact us without hesitation.
Reply to the comments on atmosphere-1745168:
1.In the manuscript the authors presented the air pollutant data obtained from automatic measuring station in Ulaanbaatar city from 2016-2019. Temporal and seasonal variations of PM10, PM2.5, SO2, NO2, CO, and O3 together with inter-annual comparison were analyzed. The correlation between pollutants is also presented. In addition, hourly variation in PM was described and for several heavy polluted days a 48-hour backward trajectory were presented indicating possible transport contribution.
The obtained results shows that the air quality in UB improved significantly in 2019 compared with that in 2016 as a result of different mitigation applied.
This is a very good written manuscript, but from my point of view rather technical. The authors use routine available measurements and descriptive statistics to show basic variations, correlations and trends. There are no new experimental/theoretical approach, but standard very good written report. Although it can be useful for local/regional air quality strategy, there is no results that could be of wider scientific interest (there are many data from the cities worldwide analyzed in the similar way through standard air quality plans/reports).
I suggest, based on the obtained results, to further analyze the database but using more robust techniques like AI or machine learning algorithm to investigate in depth the relation between pollutants and other parameters (meteorology…).
Reply: As you mentioned, we do use routinely available measures and descriptive statistics to show fundamental changes, correlations, and trends. The main reason for this is because in Mongolia (where we are located), access to experimental equipment is very limited, making it difficult for us to employ advanced research methods used by many of our peers. Few studies have used the same method as ours to analyze the air pollution in UB City. Therefore, our research can provide the scientific community with a greater understanding of Mongolia's air quality and local air pollution control. To more fully understanding the air quality status of UB City and its influencing factors. In future research, we are currently using MATLAB software to combine automatic monitoring data for ambient air quality with meteorological, aerosol, and other data to establish a predictive model of PM2.5 in UB City, which will provide data for the city's air quality governance.
2.Some minor technical corrections:
Figure 3. Percentage of days with different levels of air quality in UB from 2016 to 2019.
There is no Figure 3 in the manuscript. Please check it.
Line 178: “…and PM2.5/PM10 …” Please use subscripts
Line 196: same as previous suggestion
Reply: Thank you for your correction, we have made changes in the article.
All the lines and pages indicated above are in the revised manuscript.
Thank you and all the reviewers for the kind advice.
Sincerely,
Suriya
Department of Chemistry School of Mathematics and Natural Science Mongolian National University of Education
Ulaanbaatar 210648, Mongolia
Telephone: +976-95819724
Email: suriyaa@ hlbec.edu.cn

Round 2
Reviewer 1 Report
Dear Editor
The authors have made suggested changes in the manuscript. I recommend it published in its present form.
Author Response
Dear Reviewer:
Thank you very much for your letter dated 8 Jun. 2022, and the referees’ reports. The co-authors and I would like to thank you for the time and effort spent in reviewing the manuscript. Based on comments from you and the reviewers, we have made extensive modifications to the original manuscript. Here, we attach the revised manuscript in the format of MS Word for your approval. A document answering every question from the referees is also summarized and enclosed. Should you have any questions, please contact us without hesitation.
Reply to the comments on atmosphere-1745168:
- The reviewer’s comment: the authors have made suggested changes in the manuscript. I recommend it published in its present form.
Reply: Thank you for your recognition of our work, and good luck!
All the lines and pages indicated above are in the revised manuscript.
Thank you and all the reviewers for the kind advice.
Sincerely,
Suriya
Department of Chemistry School of Mathematics and Natural Science Mongolian National University of Education
Ulaanbaatar 210648, Mongolia
Telephone: +976-95819724
Email: suriyaa@ hlbec.edu.cn
Reviewer 3 Report
The revised version of the manuscript is slightly improved and it is a very good technical paper. I support the authors plans to implement different predictive models based on the presented data, which I believe could be of wider scientific interest in future.
Author Response
Dear Reviewer:
Thank you very much for your letter dated 8 Jun. 2022, and the referees’ reports. The co-authors and I would like to thank you for the time and effort spent in reviewing the manuscript. Based on comments from you and the reviewers, we have made extensive modifications to the original manuscript. Here, we attach the revised manuscript in the format of MS Word for your approval. A document answering every question from the referees is also summarized and enclosed. Should you have any questions, please contact us without hesitation.
Reply to the comments on atmosphere-1745168:
- The reviewer’s comment: The revised version of the manuscript is slightly improved and it is a very good technical paper. I support the authors plans to implement different predictive models based on the presented data, which I believe could be of wider scientific interest in future.
Reply: Thank you for your recognition of our work, and good luck!
All the lines and pages indicated above are in the revised manuscript.
Thank you and all the reviewers for the kind advice.
Sincerely,
Suriya
Department of Chemistry School of Mathematics and Natural Science Mongolian National University of Education
Ulaanbaatar 210648, Mongolia
Telephone: +976-95819724
Email: suriyaa@ hlbec.edu.cn